# Immunomodulatory Drugs in Acute Myeloid Leukemia Treatment

**DOI:** 10.3390/cancers12092528

**Published:** 2020-09-05

**Authors:** Antonio Piccolomo, Claudia Pia Schifone, Vanda Strafella, Giorgina Specchia, Pellegrino Musto, Francesco Albano

**Affiliations:** 1Department of Emergency and Organ Transplantation (D.E.T.O.), Hematology and Stem cell Transplantation Unit, University of Bari “Aldo Moro”, 70124 Bari, Italy; antopiccolomo@libero.it (A.P.); claudia_schifone@libero.it (C.P.S.); vandastrafella@gmail.com (V.S.); pellegrino.musto@uniba.it (P.M.); 2Former Full Professor of Hematology, University of Bari “Aldo Moro”, 70124 Bari, Italy; specchiagiorgina@gmail.com

**Keywords:** immunomodulatory drugs, acute myeloid leukemia, thalidomide, lenalidomide, pomalidomide

## Abstract

**Simple Summary:**

Immunomodulatory drugs (IMiDs) are a class of molecules composed of thalidomide and its analogs. There is a growing interest in IMiDs on the treatment of acute myeloid leukemia (AML) patients. As AML has a worse prognosis, especially in elderly patients, novel drugs such as IMiDs, as monotherapy or combination with standard induction chemotherapy or hypomethylating agents, are urgently needed. The effect of IMiDs on AML immunosurveillance seems to be the clue to understanding their clinical application in this setting. We describe the several IMiDs currently available and also future directions. Clinical trials have to find the optimal dosing and schedule and combination agents to improve response rates with the use of IMiDs and, finally, survival in patients with AML.

**Abstract:**

Immunomodulatory drugs (IMiDs) are analogs of thalidomide. They have immunomodulatory, antiangiogenic and proapoptotic properties and exert a role in regulating the tumor microenvironment. Recently IMiDs have been investigated for their pleiotropic properties and their therapeutic applications in both solid tumors (melanoma, prostate carcinoma and differentiated thyroid cancer) and hematological malignancies. Nowadays, they are applied in de novo and relapsed/refractory multiple myeloma, in myelodysplastic syndrome, in del5q syndrome with specific use of lenalidomide and B-cell lymphoma. Several studies have been conducted in the last few years to explore IMiDs possible use in acute myeloid leukemia treatment. Here we report the mechanisms of action of IMiDs in acute myeloid leukemia and their potential future therapeutic application in this disease.

## 1. Introduction

Immunomodulatory drugs (IMiDs) are a class of molecules composed of thalidomide and its analogs—lenalidomide, pomalidomide and the new molecules CC-122, CC-220, CC-885, CC-9009. They have immunomodulatory and anti-proliferative properties and are applied in the treatment of several hematologic and solid malignancies as well as in autoimmune diseases [1]. IMiDs are the backbone of multiple myeloma therapies and recently their use has been extended to B-cell lymphomas and myelodysplastic syndromes (MDS) [2]. Recent studies have clarified the IMiDs mechanism of action. They exert their pleiotropic effect by binding a single primary target, cereblon (CRBN), rather than affecting multiple molecular substrates [3,4,5,6]. CRBN is part of the CLR4 E3 ubiquitin ligase complex together with damage-binding protein-1 (DDB1), cullin 4 (Cul4A or Cul4B) and regulator of cullins (Roc1). CRBN acts as a substrate receptor and binds some proteins facilitating their ubiquitination and proteasome-dependent proteolysis (Figure 1) [7]. The downstream molecular alteration of CRBN is involved in the antiangiogenetic and immunomodulatory effects of IMiDs [3]. The increasing knowledge and interest in IMiDs therapeutic action have promoted several studies on their use in acute myeloid leukemia (AML) treatment. In this scenario, IMiDs seem to have a direct impact on AML blasts and also on the bone marrow microenvironment. Some interactions occur between leukemic cells and the immune system involved in AML pathogenesis; therefore, the tumor microenvironment makes it difficult for the immune system to act against cancer cells [8]. IMiDs affect immune surveillance against leukemia—on the one hand, they induce the activation of effector cells (natural killer T (NKT) cells, natural killer (NK) cells, γδ T cells, macrophage, cytotoxic T cells), while on the other hand, they reduce inhibitory T-cell populations (regulatory Th cells (Treg) and IL-17-producing Th cells (Th17)) [9]. Moreover, these drugs have a costimulatory effect on T and NK cells by increasing interleukin-2 (IL-2) and interferon-γ (IFN-γ) (Figure 2) [10]. NK cells can enhance the immune response against cancer cells by creating different molecular bindings. The first theory proposed to explain this mechanism was defined as the “missing-self” hypothesis—NK cells were able to recognize malignant cells that lost HLA class I ligands [11]. Today, it is well-known that the anti-tumor activity of NK cells depends on different signaling molecules—killer immunoglobulin-like receptors (KIRs), that are inhibitory receptors which recognize HLA class I ligands, natural killer group 2A (NKG2A, inhibitory as well) and innate cytotoxic receptors (NCRs, an activator family) [12]. In the AML setting, activating NK cells receptors (DNAM 1, NKG2D) are down-regulated, reducing NK cell killing. Moreover, binding between AML blasts and NK cells is ineffective because of the up-regulation of inhibitory signals; such changes have an impact on NK activity and AML prognosis [13]. Chretien et al. [14] described three NK cell maturation stages in AML patients (hypermaturation, intermediate maturation and hypomaturation); NK cells with hypomaturation features are associated with a poor prognosis. IMiDs downregulate the expression of HLA class I molecules on AML cells, increase their disruption by NK cells and raise the expression of crucial ligands (particularly ligands of DNAM-1 and NKG2D) involved in blasts recognition [12,15]. Furthermore, these drugs promote NK cell phenotypic modifications, improving the immunological synapse between NK and leukemic cells (upregulation of CD56 and downregulation of KIR2D, NKp30, NKp46 on NK cells) [12].

Finally, they enhance NK cell degranulation and the production of cytokines (IFN-γ and TNF-α) and promote NK-cell mediated antibody-dependent cell-mediated cytotoxicity against tumor cells (ADCC) [12]. It is well known that angiogenesis plays an essential role in the progression of cancer; IMiDs also have an anti-angiogenetic effect, reducing tumor necrosis factor-α (TNF-α), vascular endothelial growth factor (VEGF), fibroblast growth factor-β (FGF-β) and interleukin-6 (IL-6) (Figure 2) [16]. In the AML context, as blasts and endothelial cells are mutually dependent for survival and proliferation, therapy directed against several pro-angiogenic factors might enhance AML outcomes. Several investigations have been carried out to find new therapeutic agents for AML treatment. AML has been shown to be typical of elderly patients. In this particular subset of patients, the response to chemotherapy is inferior because of the disease’s biological characteristics and the toxicity of standard therapies [17]. Moreover, the outcome for patients with refractory or relapsed (R/R) AML is even worse and currently there is no standard therapy. IMIDs can be effective in high risk, older and R/R AML patients, as shown in several studies.

## 2. Thalidomide

The first molecule in the IMiDs family is thalidomide. Thalidomide has different mechanisms of action and has been tested in untreated and R/R AML patients. This drug has a tremendous antiangiogenic activity in AML, due to its direct inhibition of endothelial cell proliferation, reduction of fibroblast growth factor (FGF) and downregulation of a VEGF receptor, neuropilin-1, which is overexpressed in AML and whose high levels are correlated with lower survival [18]. Aguayo et al. demonstrated that VEGF levels are elevated in AML bone marrow and that the higher levels of VEGF in AML are linked to the worst prognosis [19]. These data suggest that thalidomide may play a role in treating patients with AML. Besides, thalidomide has critical immunomodulatory effects, decreasing TNF-α synthesis and selectively modulating T cell subsets [20]. Indeed this molecule shifts the T cell population towards T helpers [21]. In their phase 1/2 trial, Steins et al. [22] investigated the use of thalidomide as a single agent in AML patients who were poor candidates for intensive cytotoxic chemotherapy; these AML patients were refractory to at least two standard induction chemotherapies and were not eligible for allogeneic stem cell transplantation. In 20 evaluated AML patients, 4 patients had hematologic improvement (HI) and 4 partial response (defined as a reduction of leukemic blast infiltration of 50% in bone marrow, PR). A phase 2 trial was conducted on 16 patients with R/R AML, previously treated with a cytarabine-containing regimen [23]. They were treated daily with thalidomide 200–800 mg orally for an average of 27 days. At the end of the study, only one (6%) patient achieved complete remission (CR) lasting for 36 months; noteworthily, this patient had chromosomal aberration del(5)(q22q35). Toxicities were a limiting factor for the study—common side effects included fatigue, sedation, neurotoxicity. This study suggests that thalidomide used as a single agent is not a good therapeutic strategy for poor prognosis de novo and R/R AML. Combination strategies with thalidomide and chemotherapy have been tested; thalidomide has been associated with several agents, such as 5-azacitidine (5-AZA), amifostine, topotecan and arsenic trioxide (ATO) [21,23]. A randomized study of liposomal daunorubicin and cytarabine with or without thalidomide (400 mg escalated to 600 mg daily) in poor-risk karyotype untreated AML did not show any extension of the duration of CR with the addition of thalidomide [24]. Another phase 2 study assessed the efficacy of a quadruple regimen of thalidomide, fludarabine, carboplatin and topotecan (FCTT) in poor prognosis patients with R/R, previously treated with conventional chemotherapy or secondary AML (s-AML). Five patients achieved CR and 5 an incomplete platelet recovery (CRp), for an overall CR rate of 24%. The karyotype did not seem to be correlated with response. Moreover, the FCTT regimen, as compared to a phase 1 evaluation of fludarabine, carboplatin, topotecan regimen (FCT) alone, led to similar results—this suggested that the addition of thalidomide did not improve the effect of chemotherapy [25]. A study by Chen et al. [26] included two different arms. The first one, the control arm, received treatment with a non-intensive regimen composed of cytarabine, aclarubicin and granulocyte colony-stimulating factor (G-CSF) to induce remission; in the second arm, the investigational arm, AML patients were also given thalidomide, at a maximum dose of 200 mg/day. Seventy elderly patients were enrolled. A trend towards better overall survival (OS) and event-free survival (EFS) was observed in the investigational arm, although no statistical survival benefit was seen between the two treatment arms. The most promising data were gained with the combination of thalidomide and 5-AZA, a hypomethylating agent. Thalidomide and 5-AZA have different mechanisms of action and toxicity profiles, so they could be combined with good results. In Raza’s phase 2 study [27], the use of low-dose thalidomide in association with 5-AZA was effective in patients with AML arising from MDS. The study population comprised 40 patients with AML (de novo or post-MDS) or MDS—average age was 72 years. Eight out of 14 (57%) AML patients responded to treatment, among which 4 achieved CR. In a phase 1/2 trial, 80 patients with clinically advanced MDS, chronic myelomonocytic leukemia (CMML) and low blast count AML were treated with thalidomide and 5-AZA [28]. The eligibility criteria were—no treatment with thalidomide or its analogs in the 30 days before the study and no prior treatment with 5-AZA or another demethylating agent. Patients received 5-AZA 75 mg/mq for seven days every 28 days and thalidomide from 50 mg up to 100 mg daily; median treatment duration was 9 cycles. The combination of thalidomide and 5-AZA appeared to be more effective than 5-AZA as a single agent; CR was observed in 26% patients, PR in 5% and HI in 14%; the overall response rate (ORR) according to intention to treat (ITT) was 63%. The drugs were well tolerated; the most common side effects were infections, while 27% and 35% of patients had grade 3+ neutropenia and grade 3+ thrombocytopenia, respectively. These two studies acted as the springboard to further investigations of the association of 5-AZA with other IMiDs like lenalidomide, which is more effective than thalidomide and has a more favorable toxicity profile. The inhibitory effects of thalidomide on angiogenesis and VEGF expression in leukemic cells were emphasized in an in vitro study of a combination of thalidomide with ATO. Moreover, both ATO and thalidomide facilitate an interruption of the AML cell cycle at the G1 phase [29].

## 3. Lenalidomide

In the last few years, lenalidomide has been employed as an essential anticancer drug. It was approved by the Food and Drug Administration (FDA) to treat multiple myeloma, del5q MDS and mantle cell lymphoma. It has several mechanisms of action, including a direct anti-tumor effect, inhibition of angiogenesis, anti-inflammatory activity and immunomodulation (NK cell activation, T cell costimulation, Treg suppression). In this way, lenalidomide promotes tumor cell apoptosis both directly and indirectly, through antiangiogenic and anti-osteoclastogenic effects, immunomodulatory activity and inhibition of bone marrow stromal cell support [30]. In the last years, evidence in literature about the potential therapeutic role of lenalidomide in AML has accumulated.

### 3.1. Lenalidomide as a Single Agent

The use of lenalidomide in studies reported in literature is generally in a particular subset of AML patients—untreated elderly patients, poor cytogenetic risk or R/R patients. A meta-analysis and systematic review by Chun-Hong Xie et al. [31] reported that in AML patients treated with lenalidomide as monotherapy, the CR rate was relatively low (14%) and the ORR was 22%. In a phase 2 study by Fehniger et al. [32], untreated AML patients aged 60 years or more (median age, 71 years) received high dose (HD) lenalidomide (50 mg/day) for up to two 28-day-cycles. Thirty-three patients with intermediate (55%), unfavorable (39%) and unknown (6%) cytogenetic risk were enrolled. The overall CR/complete remission with incomplete blood count recovery (CRi) rate was 30%. Moreover, in another report, 16% of R/R AML patients treated with HD lenalidomide achieved CR/CRi [33]. Fehniger et al. [34] reported two elderly AML patients (one untreated and one already treated) treated with HD lenalidomide. They had trisomy 13 as the only cytogenetic abnormality and they both achieved CR lasting 9 months. Moreover, Lancet et al. [35] described a 55-year-old AML patient with chromosome 5q deletion, treated with lenalidomide as a single agent at a dose of 10 mg/day, who achieved a CR. Currently, the precise pathogenic role of chromosome 5q deletion, which is associated with a poor prognosis, is unknown in AML but this case provides a basis for further investigations on the use of lenalidomide in myeloid pathologies. Some phase 2 studies showed that lenalidomide as a single agent at the standard dose (5 to 25 mg daily) has limited activity in R/R AML and MDS patients with chromosome 5q deletion in the context of a complex karyotype; on the other hand, lenalidomide may be a treatment option for AML and MDS patients with this kind of chromosomal aberration in non-complex karyotype, in particular for unfit patients [36]. Relevant safety data showed that the most common side effects of lenalidomide monotherapy were myelosuppression, fatigue and electrolyte disturbance. Infection and neutropenic fever are the most frequent complications [31].

### 3.2. Lenalidomide and Chemotherapy

The most crucial lenalidomide associations reported are those with cytarabine, clofarabine and therapeutic regimens such as mitoxantrone, etoposide, cytarabine (MEC) and cytarabine plus daunorubicin (“3+7” regimen). De Angelo. et al. [37] enrolled 35 R/R AML patients aged 18–70 years to evaluate the safety and tolerability of lenalidomide with MEC. The lenalidomide dose was escalated, starting from 5 mg increased up to 50 mg. The maximum tolerated dose (MTD) of lenalidomide was 50 mg/day in days 1–10. In this study, the safety profile was similar to events reported in other lenalidomide and MEC trials. The median OS for all patients was 11.5 months; the CR rate was 34% while, historically, the CR in MEC reinduction therapy was approximately 18–26% [38,39]. Jain et al. [40] described the association of lenalidomide and clofarabine in 4 patients with high-risk MDS and R/R AML, ineligible for intensive chemotherapy or allogeneic stem cell transplantation. Two subjects achieved stable disease and one achieved a partial response. However, all patients had to be removed from the study because of disease progression (3 cases) and liver toxicity (1 case). Despite the limited sample size of this study, it led to an exciting finding underlined by the authors—pre-treatment samples showed elevated expression of T cells exhaustion markers (PD1, LAG3 and TIM3), which are known to produce impairment of T-cells. The Authors also found that NK cells had a higher expression of LIR1, an inhibitory molecule known to induce inability of leukemia killing. The hypothesis is that in responders and stable disease subjects, clofarabine may produce lymphodepletion and a decreased number of exhausted T CD4+ and NK cells with inhibitory markers. These circumstances could create a favorable microenvironment for subsequent lenalidomide therapy, stimulating NK and T cell reconstitution. The study by Ades et al. [41] reported the association of classical “3+7” with an escalating dose of lenalidomide. The authors enrolled 82 patients with MDS or AML with 5q deletion (62 with AML). Sixty-two (76%) patients had a complex karyotype—46% of patients achieved CR and the overall response rate (ORR) was 58.5%; the CR rate in AML was a little lower (40%). The 1-year cumulative incidence of relapse was 64.6% and the median overall survival (OS) was 8.2 months. Compared with conventional intensive chemotherapy, the treatment protocol used by the authors produced higher hematologic CR rates in patients with a very poor cytogenetics risk but the response duration was short. Moreover, the Dutch-Belgian Hemato-Oncology Cooperative Group (HOVON) and the Swiss Group for Clinical Cancer Research (SAKK) conducted a randomized phase 2 study [42] with or without lenalidomide at a dose of 20 mg/day 1–21. In the second cycle, patients received cytarabine 1000 mg/m^2^ twice daily on days 1–6 with or without lenalidomide (20 mg/day 1–21). The enrolled patients were 222 previously untreated with newly diagnosed AML or high-risk MDS (aged 66 years or older). The study results showed that the addition of lenalidomide did not improve the therapeutic outcome of older AML patients. The CR/CRi rates in the two arms were not different (69 vs. 66%), the EFS at 36 months was 19% for the standard versus 21% for the lenalidomide arm and OS was 35% vs. 30%, respectively. The lenalidomide/cytarabine association was investigated in different AML settings, yielding controversial results. Griffiths et al. [43] enrolled 32 patients with R/R AML. The MTD for this association was 10 mg/daily administered on days 6–26 in a 28-day cycle. The CR/CRi rate was about 30% and median OS was 5.8 months. This trial was associated with marked skin toxicity and other toxicities such as nausea, vomiting, mucositis, electrolyte disturbances. In a second study, Visani et al. [44] described 66 elderly AML patients (median age 76 years) ineligible for intensive chemotherapy or allotransplantation, who were consecutively treated with low dose lenalidomide (10 mg/day, days 1–21) plus low dose cytarabine (10 mg/m2, twice a day, on days 1–15, every six weeks, up to 6 cycles). The rate of CR was 36% and its achievement was not predicted by bone marrow blast count, molecular markers, cytogenetics, white blood cell count or prior MDS. The main toxicities were hematological and included thrombocytopenia, anemia and neutropenic fever. It has also been reported 31 elderly AML patients with unfavorable (52%) and intermediate (48%) karyotypes treated with low dose lenalidomide (10 mg/day, days 1–21) plus low dose cytarabine (20 mg/m^2^, twice daily, days 1–15). The CR rate was 33%. In this study, non-hematological toxicities were mild [45].

### 3.3. Lenalidomide and 5-AZA

Pollyea et al. [46] evaluated the efficacy and safety of sequential 5-AZA plus lenalidomide for elderly patients with untreated AML. They enrolled 42 AML patients aged over 60 years, who were given 5-AZA (75 mg/m^2^ for seven days) followed by an escalating dose of lenalidomide daily for 21 days on day 8 of each cycle, for six weeks. The ORR was 40%, the median duration of response was 28 weeks and the OS for responders was 20 weeks. The lenalidomide MTD was 50 mg/daily. Patients with adverse risk in terms of genomic abnormalities had similar responses to those with lower risk. These findings suggested that unknown gene alterations could cause treatment sensitivity in such patients. The results indicated that sequential administration of lenalidomide and 5-AZA in untreated elderly AML patients as a treatment option is preferable to a single-drug treatment because there is a higher ORR and it is relatively well tolerated. Ramsingh et al. [47], in a phase 1 study, partly confirmed the findings by Pollyea et al. [46]. In their study, the authors enrolled 19 newly diagnosed or R/R AML elderly patients (median age 72 years) treated simultaneously with HD lenalidomide and 5-AZA. Thirteen (68%) out of 19 patients completed at least one induction cycle and 4 (30.8%) out of 13 evaluable patients obtained a CR/CRi. In a systematic review and meta-analysis by Xie et al. [31], some studies on lenalidomide and 5-AZA in AML patients were considered. The analysis revealed that the CR rate was 22% and the ORR was 31%. In this meta-analysis, the authors found that the cytogenetic risk had a minimal impact on the patient outcomes among those who received a combination of lenalidomide and 5-AZA. Myelosuppression was the most common toxicity, while infection and neutropenic fever were the most frequent complications. Kunacheewa et al. [48] evaluated the efficacy and adverse effects of 5-AZA plus lenalidomide in patients with AML, MDS and CMML. In this meta-analysis, the authors considered 10 studies for a total of 406 patients treated with a lenalidomide-plus-5-AZA regimen. The pooled CR rate was 33%, while the pooled ORR was 49%. Analysis of the subgroups revealed that patients with 5q deletion had a higher CR rate (43.8%). The most common adverse events included—grade 3–4 neutropenia (48.8%), platelet toxicity (54.7%) and febrile neutropenia (36.7%), while the most frequent non-hematological toxicities were the acute renal failure (9.2%) and thrombotic events (5.3%). Therefore, there is considerable evidence in the literature of positive results following the association of lenalidomide and 5-AZA in AML. Which specific AML population could benefit from the association of these drugs should be better defined.

### 3.4. Lenalidomide and Allogeneic Stem Cell Transplantation

The use of lenalidomide-based maintenance post allotransplantation was evaluated in the phase 2 trial LENAMAINT [49]. Ten patients were recruited—1 patient with high-risk MDS and 9 patients with AML, with a median age of 65, in CR after hematopoietic stem cell transplantation (HSCT). In this trial, lenalidomide maintenance (10 mg/day for 21 days of a 28-day cycle) was suspected of inducing acute graft-versus-host disease (aGVHD) in 6 (60%) out of 10 patients. Therefore, the use of a lenalidomide-based maintenance therapy after allogeneic HSCT does not seem to be the right choice for AML patients. However, more promising results were derived from the association with 5-AZA. In the VIOLA phase 1 trial [50], 29 patients with AML or MDS, relapsed after HSCT, were treated with 5-AZA and lenalidomide for a median follow-up of 23 months. A significant clinical response was achieved in 7 (47%) out of 15 patients. In this trial, GVHD appears less frequently as an adverse effect, probably because of the association with 5-AZA that favors Treg cell expansion, as observed in mouse models [51].

## 4. Pomalidomide

Pomalidomide is a third-generation immunomodulatory drug approved for the treatment of R/R multiple myeloma. Like the other molecules previously described, pomalidomide has anticancer, antiangiogenic and immunomodulatory properties. It seems to enhance anti-tumor immune response costimulating T cells, increasing both the activity and the proliferation of NK and NKT cells. In contrast to thalidomide, it reduces the levels of Treg and Th17 cells [9]. Recent studies have investigated pomalidomide’s role in AML treatment providing limited evidence but underlining its action on NK and T cells. In “in vitro” and murine models, pomalidomide has a direct and indirect activity against AML blasts, reducing their vitality and sensitizing leukemic cells to NK. As explained above, IMiDs, mainly pomalidomide, lower HLA-I class molecule expression on AML blasts, increase CD56 expression and downregulate the expression of the inhibitory receptor (KIR2D) on NK cells. Similarly to lenalidomide, this drug enhances NK degranulation, improving the polarization of this immunological synapse [12]. The binding between CRBN and pomalidomide seems to be involved in immunomodulatory effects. Pomalidomide enhances ubiquitination and degradation of Ikaros and Aiolos, two transcription factors expressed in several hematological diseases and crucial for the development of lymphoid lineage and lymphocytes functions (Figure 1). Aiolos is also expressed in AML; its role is not yet entirely understood. Gandhi et al. [52] reported that Aiolos is an IL-2 promoter repressor; consequently, its degradation, induced by pomalidomide, facilitates IL-2 costimulation of CD4+ Th1 lymphocytes. It will be necessary to clarify the role of Aiolos in AML and its action on T cells populations and interleukins production. In a phase 1 trial (NCT02029950), Zeidner et al. [53] enrolled 51 AML patients (46 de novo non-favorable risk AML, 4 high-risk MDS and 1 CMML-2) who received timed sequential therapy (TST) as induction treatment (AcDVP-16 regimen, cytarabine, etoposide, daunorubicin), pomalidomide escalating doses at early lymphocyte recovery (ELR) and as maintenance therapy. The first objective was to establish the pomalidomide MTD—4 mg/day for 21 days of a 28-day cycle; additionally, the trial demonstrated an improvement in OS and EFS in pomalidomide-treated high-risk AML. This study also reported preliminary results regarding pomalidomide immunomodulatory activity. During the immunomodulant administration, the Aiolos concentration was reduced in CD4+ and CD8+ T-lymphocytes from bone marrow (BM) and peripheral blood (PB) samples. IL-2 and TNF-α production increased in BM CD8+ and PB CD4+; similarly, there was a growth in the frequency of effector memory lymphocytes T subpopulation. Gene expression analysis reveals an upregulation of genes involved in metabolic and immune processes in CD8+ and CD4+ associated with a downregulation of exhaustion genes as well as Treg genes. Further studies are necessary to define the effect of pomalidomide on ELR, T lymphocytes and NK subpopulations and to clarify the impact of immunological modifications on both AML immune surveillance and disease outcome.

## 5. New Molecules

New thalidomide analogs have recently been developed and are currently involved in different clinical trials to define their mechanisms of action. The effects of the new molecules depend on their binding to CRBN. CC-122 (iberdomide) and CC-220 (avodamide) enhance Aiolos and Ikaros degradation and are under investigation for their possible role in lymphoproliferative and rheumatologic disorders [54]. CC-885 inhibits cancer growth and it has been tested against different cell lines and AML leukemic blasts isolated from patients samples. Mainly, CC-885 facilitates the binding of GSPT1 (G1 to S phase transition 1, a translation termination factor) to CRBN, enhancing the degradation of this protein (Figure 1). GSPT1 binds to eukaryotic translation termination factor 1(eRF1) and the deriving complex recognizes proteins’ stop codons, allowing their release from ribosomes. The inhibition of this process reduces cellular fitness and seems to have an anti-proliferative effect [55]. CC-9009 enhances the binding of Ikaros and Aiolos to CRBN [10]. In preclinical studies on AML cell lines, CC-9009 binds to GSPT1 inducing apoptosis of AML blasts. CC-9009 has been applied in a phase 1 clinical trial in vivo to evaluate its pharmacodynamics, pharmacokinetics and tolerability in heavily pretreated R/R AML patients. In this setting, the new drug seems to improve the outcome and reduce the GSPT1 concentration in leukemic blasts and T cells from patients’ PB samples [56]. The CLR4 E3 ubiquitin ligase complex in IMiDs action highlights its role in protein ubiquitination. Proteolysis targeting chimeras (PROTACs) is composed of two molecules binding a target protein with an E3 ubiquitin ligase, promoting target intracellular ubiquitination. dBET1 is a PROTAC, composed of thalidomide and JQ1; the latter binds bromodomain and extra-terminal motif transcription factor (BET, specifically BRD4) promoting its degradation [57]. BRD4 is a chromatin reader protein that acts as a transcriptional coactivator involved in physiological hematopoiesis and is recruited in cancer development. In AML, this protein promotes chromatin remodeling and the transcription of pivotal genes for the tumor survival. Remarkably, BRD4 inhibition seems to specifically affect the transcription of cancer-promoting genes [58]. ARV-285 is composed of pomalidomide and OTX015, a BRD4 inhibitor. A preliminary study conducted on AML cell lines and cells derived from s-AML patients (post-Myeloproliferative Neoplasms) indicates that PROTAC brings about a profound disruption of s-AML oncogenes through BRD4 degradation [59]. These small molecules promote the CRBN-mediated ubiquitination of BRD4 and are thus possible candidates for AML treatment.

## 6. Conclusions

In this review, we focus on the IMiDs pleiotropic action in the AML scenario (in Table 1 are reported the AML clinical studies with new IMiDs). The majority of clinical trials have been focused on thalidomide and lenalidomide applied as single agents or in combination therapy. The results are impressive but not wholly conclusive. Thalidomide exerts its immunomodulatory effect through the costimulation of T cells (partially activated by their T-cell receptor) and enhancement of NK cells cytotoxicity. On the other hand, thalidomide is less active against Treg proliferation [9]. The Tregs level is high in AML patients at diagnosis and ELR after TST induction therapy; they constitute an independent prognostic factor and facilitate immunosurveillance escape [60]. Moreover, thalidomide binds CRBN directly and DDB1 (two compounds of CLR4 E3 ubiquitin ligase complex) indirectly but this binding is less stable than those produced by lenalidomide and pomalidomide and even Ikaros and Aiolos degradation is weak [3,6]. Thus, thalidomide may be less effective than its more recently developed analogs in activating the immune system against leukemic blasts. Thalidomide’s clinical trials often involve cohorts composed of heavily pretreated AML patients, probably affecting clinical results and drug impact on the immune system. In the studies examined, this drug used as a single agent did not prove effective; toxicities were often significant, especially at high doses and precluded dose escalation and study continuance. The association of thalidomide with hypomethylating drugs like 5-AZA in trials is attractive and was the starting point for new further studies with its derivative drug, lenalidomide. Lenalidomide is the most frequently investigated molecule in clinical trials probing the role of IMiDs in AML. This drug has a better toxicity profile and efficacy than thalidomide. Previous studies provided some evidence about the effects of lenalidomide in AML treatment. Probably, the most significant results occur following two combination therapies—lenalidomide plus cytarabine and lenalidomide plus 5-AZA, suggesting a possible synergic action of these molecules. The most important studies reported a response rate of approximately 30% for these two combinations [31]. It was shown that the cytogenetic risk had exerted only a minimal impact on the outcomes of lenalidomide-based regimens.

Finally, it may be speculated that there is a particular subset of AML patients who could benefit from these treatments—elderly patients and R/R who are not eligible for intensive treatment. Undoubtedly, further studies and evidence are required to better define the target population to be treated with this kind of drug association. Regarding safety, lenalidomide-based regimens were well tolerated but the association with cytarabine or 5-AZA can increase toxicities. It is noteworthy that lenalidomide has a remarkable impact on the treatment of MDS associated with del(5q) as a single chromosomal alteration because it induces casein kinase 1 (Ck1α) degradation. This protein is encoded by a gene mapped on chromosome 5q and lenalidomide enhances its reduction allowing p53 activation (Figure 1) [3,61]. Järås et al. [62] showed that in AML, Ck1α partial inhibition promotes p53 activation and leukemic blast elimination, highlighting its crucial role for tumor survival and potential importance as a therapeutic target. Hypomethylating agents downregulate NKG2D inhibitory ligands production by AML cells, improving NK activity [15]. This evidence might at least partially explain the potential synergic action with lenalidomide. The impact of the combination of cytarabine with lenalidomide deserves consideration. The reported studies showed an improvement in the AML outcome—cytarabine cytotoxicity seems to act synergistically with lenalidomide’s impact on the microenvironment to disrupt leukemic blasts.

As far as the use of pomalidomide and the new molecules in AML is concerned, studies are still few but promising. Several studies have highlighted the relevance of NK and T cells in AML development but it is necessary to clarify the role of these cells as therapeutic targets. The effect of IMiDs on AML immunosurveillance seems to be the clue to understanding their clinical application in this setting. The discovery that interactions with the cells of the microenvironment play an essential role in activating processes of the proliferation of leukemic cells is also of interest. The central role of CRBN binding in the wide range of mechanisms and targets of action of IMiDs has raised notable interest in their application in AML patients; certainly, further studies will be required to establish the therapeutic use of these drugs in AML. Since 2017, there has been an increase of newly approved AML treatment options, with the majority of new drugs targeting specific gene mutations (such as midostaurin, enasidenib and ivosidenib) and pivotal cell survival pathways (such as venetoclax) [63]. At present, there are not enough data to know how best to use these newly approved drugs in terms of combination and sequential therapy. Combinations of these new agents with IMiDs could be in the near future an AML therapeutic approach worthy of investigation.

## Figures and Tables

**Figure 1 cancers-12-02528-f001:**
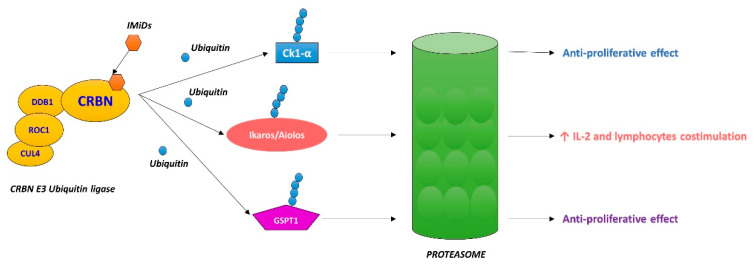
Cereblon (CRBN), a single primary target of immunomodulatory drugs (IMiDs). IMiDs bind CRBN, enhancing the ubiquitination and the subsequent proteolysis of different transcription factors. The degradation of Aiolos and Ikaros facilitates lymphocytes stimulation and their anticancer activity; the lysis of Ck-1α and GSPT1 inhibits malignant cell proliferation.

**Figure 2 cancers-12-02528-f002:**
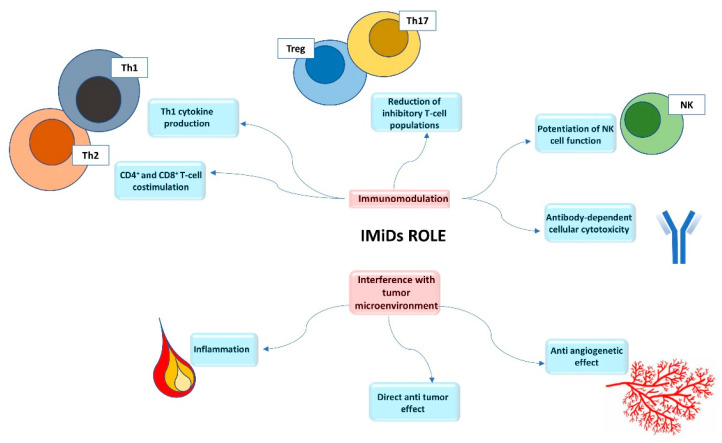
The role of IMiDs. IMiDs act in two different ways: they have an immunomodulating role and they also interfere with the tumor microenvironment. As regards immune modulation, the main actions are CD4^+^ and CD8^+^ T-cell costimulation, Th1 cytokine production, the reduction of inhibitory T-cell populations and boosting of NK cell function, antibody-dependent cellular cytotoxicity. IMiDs interfere with the tumor microenvironment: they have an antiangiogenetic role, anti-inflammatory properties and a direct anti-tumor effect.

**Table 1 cancers-12-02528-t001:** New IMiDs in acute myeloid leukemia (AML) clinical studies.

Clinical Study.	AML Patients (n)	Treatment	Conclusion
Fehniger et al. [32]	Untreated AML (33)	HD lenalidomide	The overall CR/Cri was 30%
Blum et al. [33]	R/R AML (31)	HD lenalidomide	The overall CR/Cri was 16%
Chen et al. [36]	R/R AML (18)	Lenalidomide (standard dose)	Clinical activity of lenalidomide as a single agent in AML with chromosome 5 abnormalities appeared to be limited to patients with noncomplex cytogenetics
De Angelo et al. [37]	R/R AML (35)	MEC + lenalidomide	The combination of lenalidomide and MEC chemotherapy was well-tolerated and efficacy, even among patients with highly resistant disease
Ades et al. [41]	Untreated AML with del(5q) (62)	“3+7” plus lenalidomide vs. “3+7”	Compared with standard “3+7”, the treatment including lenalidomide produced higher hematologic CR rates in patients with a very poor cytogenetics risk but the response duration was short
Ossenkoppele et al. [42]	Untreated AML (200)	“3+7” plus lenalidomide vs. “3+7”	The addition of lenalidomide to standard induction does not improve the therapeutic outcome of older AML patients
Griffiths et al. [43]	R/R AML (32)	Lenalidomide + cytarabine	The combination did not appear to result in improved CR over single-agent cytarabine for R/R AML
Visani et al. [44]	Untreated AML (66)	Lenalidomide + cytarabine	The CR rate was 36%
Visani et al. [45]	Untreated AML (31)	Lenalidomide + cytarabine	The CR rate was 33%
Pollyea et al. [46]	Untreated AML (42)	5-AZA + lenalidomide	The results indicated that sequential administration of lenalidomide and 5-AZA as a treatment option is preferable to a single-drug treatment because there is a higher ORR
Ramsingh et al. [47]	Untreated AML (9) R/R AML (10)	HD lenalidomide + 5-AZA	This combination was well tolerated
Sockel et al. [49]	AML with del(5q) in CR after allogeneic HSCT (9)	Lenalidomide maintenance after allogeneic HSCT	Lenalidomide maintenance to prevent relapse following allogeneic HSCT is not feasible, mainly due to the induction of severe aGvHD
Craddock et al. [50]	Relapsed AML after allo-SCT (24)	5-AZA + lenalidomide	This combination may have a potentially important role as salvage therapy in patients with relapsed AML post-allograft
Zeidner et al. [53]	Untreated AML (46)	Cytarabine + daunorubicin + etoposide induction therapy followed by pomalidomide	The CR/Cri rate was 86%
Uy et al. [56]	R/R AML (45)	CC-9009	In this phase 1 study of CC-90009 a promising antileukemic activity was observed

AML: acute myeloid leukemia; HD: high dose; CR/Cri: complete remission/complete remission with incomplete blood count recovery; R/R: refractory or relapsed; MEC: mitoxantrone, etoposide, cytarabine; “3+7”: cytarabine plus daunorubicin; 5-AZA: 5-azacitidine; ORR: overall response rate; HSCT: hematopoietic stem cell transplantation; aGvHD: acute graft-versus-host disease.

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
