# Peer review of "Immunomodulatory Drugs in Acute Myeloid Leukemia Treatment"

_cancers, 2020, doi:10.3390/cancers12092528_

Round 1

Reviewer 1 Report

The article by Antonio Piccolomo et al. is a well-written review article about the role and mode of action of immunomodulatory drugs (IMiDs) in acute myeloid leukemia). As AML has a worse prognosis, especially in elderly patients, novel drugs, as monotherapy or combination with induction chemotherapy or hypomethylating agents, are urgent needed. The article describes the several IMiDs currently available and also future directions. The overview is complete the cited references actually. The several mode of action of IMiDs are described in detail: immunomodulatory, anti-angiogenic and also the target cereblon and its downstream signalling. All relevant clinical studies with the use of IMids, either as monotherapy or In combination, are mentioned. In conclusion, the review article by Antonio Piccolomo et al. gives a comprehensive overview of the mode of action of IMIds in AML and the currently available clinical data. In the future, clinical trials have to find the optimal dosing and schedule, and combination agents, to improve response rates with the use of IMids and finally survival in patients with AML.

Author Response

No revision was requested

Reviewer 2 Report

Thank you for this comprehensive and well written review on the use of immunomodulatory drugs in the treatment of acute myeloid leukemia. The review is well organized and presents each of the different IMiD drugs categorically. 

There is some mixing in of studies of AML patiernts with myelodysplasia and chronic myelomonocytic monocytic leukemia patients; however I feel the authors tried to discriminate when those situations arose and commented where possible to delineate the impact on AML patients.

I do not feel you tried to overstate the impact of the ImiDs in the treatment of AML and proposed appropriately that further study is warranted.

Author Response

No revision was requested

Reviewer 3 Report

The authors summarized the action of immunomodulatory (IMiDs) drugs in acute myeloid leukemia. They described conventional IMiDs and next generation IMiDs. The review article is clear presented and well organized. It is easy to read and very focused on the title. However some improvements can be made. My comments are below:

1) Figure 1 and 2 resolution need to be improved. The font of the single component binding to CRBN is very small (almost illegible). I suggest to increase the font of at least 3 sizes.

2) A Table with description of clinical trials for the new IMiDs should be included. It will be easier for the readers and will provide a quick guide while reading.

3) A small paragraph highlighting basic experimental studies for the conventional IMiDs could be included since the title of the review is general. The addition of a small paragraph will help summarizing better the Figure 1 that the authors have included.
